# Design, Synthesis, and Evaluation of a Set of Carboxylic Acid and Phosphate Prodrugs Derived from HBV Capsid Protein Allosteric Modulator NVR 3-778

**DOI:** 10.3390/molecules27185987

**Published:** 2022-09-14

**Authors:** Xiangkai Ji, Xiangyi Jiang, Chisa Kobayashi, Yujie Ren, Lide Hu, Zhen Gao, Dongwei Kang, Ruifang Jia, Xujie Zhang, Shujie Zhao, Koichi Watashi, Xinyong Liu, Peng Zhan

**Affiliations:** 1Department of Medicinal Chemistry, Key Laboratory of Chemical Biology (Ministry of Education), School of Pharmaceutical Sciences, Cheeloo College of Medicine, Shandong University, 44 West Culture Road, Jinan 250012, China; 2Department of Virology II, National Institute of Infectious Diseases, Tokyo 163-8001, Japan; 3Department of Applied Biological Science, Tokyo University of Science, Noda 278-8510, Japan; 4Research Center for Drug and Vaccine Development, National Institute of Infectious Diseases, Tokyo 163-8001, Japan

**Keywords:** HBV, capsid, CpAM, NVR 3-778, prodrug strategy, water solubility

## Abstract

Hepatitis B virus (HBV) capsid protein (Cp) is necessary for viral replication and the maintenance of viral persistence, having become an attractive target of anti-HBV drugs. To improve the water solubility of HBV capsid protein allosteric modulator (CpAM) NVR 3-778, a series of novel carboxylic acid and phosphate prodrugs were designed and synthesized using a prodrug strategy. In vitro HBV replication assay showed that these prodrugs maintained favorable antiviral potency (EC_50_ = 0.28–0.42 µM), which was comparable to that of NVR 3-778 (EC_50_ = 0.38 µM). More importantly, the cytotoxicity of prodrug **N8** (CC_50_ > 256 µM) was significantly reduced compared to NVR 3-778 (CC_50_ = 13.65 ± 0.21 µM). In addition, the water solubility of prodrug **N6** was hundreds of times better than that of NVR 3-778 in three phosphate buffers with various pH levels (2.0, 7.0, 7.4). In addition, **N6** demonstrated excellent plasma and blood stability in vitro and good pharmacokinetic properties in rats. Finally, the hemisuccinate prodrug **N6** significantly improved the candidate drug NVR 3-778’s water solubility and increased metabolic stability while maintaining its antiviral efficacy.

## 1. Introduction

Chronic hepatitis B is an infectious disease characterized by liver damage caused by persistent infection with hepatitis B virus (HBV). Further deterioration of hepatitis B will cause a series of complications, such as disorders of liver metabolism, liver failure, cirrhosis, and liver cancer [1]. According to the World Health Organization, about 296 million people worldwide suffer from chronic HBV infection, accounting for about 4% of the world’s population, and about 820,000 people die of chronic viral-hepatitis-related liver disease every year [2]. Hepatitis B is highly contagious and difficult to cure, which seriously endangers human public health and social development. Therefore, it is urgent to develop safe and effective anti-HBV drugs.

At present, interferons and nucleos(t)ide analogues are used to treat HBV but neither can achieve elimination of HBV. The goal in the treatment of hepatitis B is to maximize the inhibition of HBV replication, reduce the HBV antigen levels, relieve the symptoms of hepatitis and liver fibrosis, reduce and delay the occurrence of complications, thereby improving the liver function and quality of life of patients [3]. Clinically, long-term use of the current drugs accelerates the occurrence of drug resistance or adverse reactions. In addition, nucleos(t)ide analogues have to be used for a prolonged period or possibly life-long to continuously inhibit HBV replication [4,5,6]. Therefore, it becomes a hot area of research in anti-HBV drugs that use new strategies of medicinal chemistry to develop non-nucleoside HBV inhibitors with novel mechanisms of action.

HBV capsid protein (Cp) plays a key role in multiple stages of HBV replication, including subcellular transport, cccDNA maintenance, capsid assembly, and the following processes of pregeomic RNA encapsidation and viral DNA synthesis [7,8,9]. Therefore, HBV Cp has become an attractive target for mechanism-oriented antiviral therapy due to the important role in viral replication. Capsid protein allosteric modulators (CpAMs) disrupt the functional capsid assembly by directly targeting Cp, thereby inhibiting HBV replication [10]. The CpAMs reported so far include heteroaryl-dihydropyrimidines (HAPs), sulfamoylbenzamides (SBAs), and phenylacrylamides (PPAs) [11,12,13]. The representative compounds in different structural types are shown in Figure 1.

NVR 3-778 (**2**) is an SBAs-CpAM that was demonstrated to have antiviral activity in HBV-infected patients [11]. NVR 3-778 showed significant antiviral activity in the HepG2.2.15 cell line (EC_50_ = 0.40 µM). It has synergistic antiviral effects with nucleoside inhibitors in vitro [14] and with polyethylene glycol interferon in clinical trials [15]. However, the relatively low water solubility (0.003 mg/mL at pH = 2.0) could limit its clinical application, and its development was stopped because of a limited efficacy at clinically feasible doses [16].

Many drugs with excellent pharmacological activity often had defects in physical characteristics and pharmacokinetics, such as poor water solubility, low oral bioavailability, and rapid metabolism, which limited the direct clinical application [17,18,19]. Both carboxylic acid and phosphate are charged groups, and the introduction of such groups can improve the water solubility of compounds. For example, after the introduction of carboxylic acid, the water solubility of adrenocorticotropic hormone prednisolone was improved and can be used for injection. In addition, when the antifungal drug fluconazole was introduced to the phosphate, its water solubility was improved and the dosage was reduced [20,21,22]. Consequently, in this study, to improve the water solubility and reduce the toxicity of NVR 3-778, we designed and synthesized a series of carboxylic acid and phosphate prodrugs.

Herein, four carboxylic acid and phosphate compounds were tested for anti-HBV activity in vitro. Among them, **N6** was selected for further water solubility testing, plasma and whole blood stability experiments, and preliminary pharmacokinetic evaluation in rats. These experimental results will be discussed in detail.

## 2. Results and Discussion

### 2.1. Synthesis of the Target Molecules

As shown in Figure 1, 3-chlorosulfonyl-4-fluorobenzoic acid (**N1**) was reacted with 4-hydroxypiperidine to obtain intermediate **N2** by nucleophilic sulfonyl substitution reaction. **N2** was treated with 3,4,5-trifluoroaniline to yield NVR 3-778 (**N3**) by the amide condensation reaction [23]. Next, **N3** reacted with various anhydrides to give end-product prodrugs **N6**–**N8** under DMAP catalysis. In addition, **N3** and dibenzyl diisopropylphosphoramidite reacted in the presence of 1H-imidazole and 3-chlorobenzoic acid to obtain **N4**. Subsequently, the benzyl group was removed from N4 under the action of hydrogen and palladium carbon to obtain phosphate prodrug **N5** [24].

### 2.2. Biological Activity

The target compounds **N5-N8** were evaluated for their antiviral potency against HBV replication, as well as cytotoxicity in Hep38.7-Tet cells, which reproduces HBV replication from a chromosome-integrated HBV transgene under depletion of tetracycline from the medium [25]. The 50% effective concentration (EC_50_) for anti-HBV activity, 50% cytotoxic concentration (CC_50_), and selectivity index (SI) calculated by CC_50_/EC_50_ ratio are shown in Table 1. The antiviral activity of all prodrugs (EC_50_ = 0.42–0.28 µM) was similar to the NVR 3-778 (EC_50_ = 0.38 ± 0.047 µM), but the toxicity of compound **N8** (CC_50_ > 256 µM, SI > 731) was reduced by at least 18-fold, compared with NVR 3-778 (CC_50_ = 13.65 ± 0.21 μM, SI = 36). When R was replaced by phosphate and 4-oxo-butanoic acid, compound **N5** (EC_50_ = 0.42 ± 0.069 µM, 13.66 ± 1.45 µM) and **N6** (EC_50_ = 0.35 ± 0.048 µM, CC_50_ = 12.62 ± 1.05 µM) had similar antiviral activity and toxicity with NVR 3-778 (EC_50_ = 0.38 ± 0.047 µM, 12.65 ± 0.21 µM). The antiviral activity of **N7** (EC_50_ = 0.28 ± 0.024 µM, CC_50_ = 12.98 ± 1.74 µM) was slightly better than that of NVR 3-788 when R was replaced by 5-oxo-butanoic acid. When R was 6-oxo-hexanoic acid, prodrug **N8** (EC_50_ = 0.35 ± 0.032 µM, CC_50_ > 256 µM) not only maintained the antiviral activity, but also greatly reduced the cytotoxicity.

### 2.3. Water Solubility

In order to verify whether the prodrug strategy was successful in improving the water solubility of NVR 3-778, representative compound **N6** was selected to test water solubility under three different pH. As shown in Table 2, compared with NVR 3-778, the water solubility of **N6** was more than 180 times at pH = 2 (**N6**: 695 µg/mL, NVR 3-778: 3.82 µg/mL), 310 times at pH = 7 (**N6**: 7500 µg/mL, NVR 3-778: 24.02 µg/mL), and 900 times at pH = 7.4 (**N6**: 4416 µg/mL, NVR 3-778: 4.84 µg/mL). The experimental data confirmed that the introduction of carboxylic acid markedly improved the solubility of NVR 3-778. The experimental results verified the rationality of prodrug strategy, and the improvement of water solubility is of great significance for the development of new pharmaceutical dosage forms.

### 2.4. Plasma and Whole Blood Stability

In order to verify the stability of prodrug in vitro, the stability of NVR 3-778 and prodrug **N6** in plasma at 4 °C and whole blood at 37 °C was tested by the LC-MS/MS and characterization of their stability by peak area. As shown in Table 3, two hours later, the residual amounts of NVR 3-778 in plasma and whole blood were 85% and 91%, respectively, and only about 10% of drugs were metabolized after 2 h of storage. Prodrug **N6** was more stable, and the residual amounts of **N6** in plasma and whole blood were close to 100%, which means almost no metabolic degradation occurred after two hours of storage. The results showed that the introduction of carboxylic acid appropriately enhanced the metabolic stability of NVR 3-778 in vitro.

### 2.5. Pharmacokinetic Experiments of ***N6***

In order to explore the metabolism and bioavailability of prodrug in vivo after oral administration, the main pharmacokinetic parameters were tested. After oral administration of **N6** and NVR 3-778 to the two groups of rats, we detected their plasma concentration, respectively. As shown in Figure 2, after oral administration of **N6**, the concentration of **N6** in plasma decreased rapidly, and **N6** could not be detected after 2 h, while the concentration of NVR 3-778 in plasma increased rapidly and reached the peak at 1 h. This indicated that **N6** was completely converted into NVR 3-778 within 2 h in rats. As shown in Table 4, compared with direct oral administration of NVR 3-778, oral prodrug **N6** (*t*_1/2_ = 1.92 ± 0.227 h) can increase the half-life of NVR 3-778 (*t*_1/2_ = 1.40 ± 0.250 h). In addition, **N6** (C_max_ = 747 ± 104 ng/mL) reduced the maximum concentration of the drug in the body compared with NVR 3-778 (C_max_ = 1161 ± 261 ng/mL), thereby appropriately reducing the side effects and irritation of drugs. Considering the different molar doses (**N6**: 18.8 nmol/kg; NVR 3-778: 23.1 nmol/kg), **N6** (AUC_0-inf_ = 2222 ± 935 ng.h/mL) had a similar drug–time curve area as NVR 3-778 (AUC_0-inf_ = 3937 ± 948 ng.h/mL), suggesting that **N6** and NVR 3-778 had analogous oral bioavailability. The experimental results confirmed that **N6** had longer half-life and lower maximum concentration in vivo than NVR 3-778. In addition, **N6** was completely transformed into NVR 3-778 in vivo, and both had favorable bioavailability. Compared with NVR3-778, **N6** can release drugs more slowly, which can improve the narrow therapeutic window of NVR3-778 to some extent.

## 3. Conclusions

To improve the water solubility and safety index of HBV CpAM NVR 3-778, we designed and synthesized a series of carboxylic acid and phosphate prodrugs. The antiviral activity of all prodrugs was similar to NVR 3-778. Compared with NVR 3-778, the cytotoxicity of compound **N8** was significantly reduced (CC_50_ > 256 µM, SI > 731). In three different pH phosphate buffer solutions (2.0, 7.0, and 7.4), the solubility of prodrug **N6** was hundreds of times better than that of NVR 3-778, which is of great significance for the development of new preparation types. In addition, **N6** exhibited excellent stability in plasma and whole blood, which provided the basis for further evaluation of druggability in vivo. Pharmacokinetics in rats showed that prodrug **N6** and NVR 3-778 had similar bioavailability. What is more, **N6** can be gradually metabolized to NVR 3-778, which reduces irritation by reducing the maximum drug concentration in vivo. In conclusion, the prodrug **N6** successfully improved the water solubility of the candidate drug NVR 3-778. While maintaining the curative effect, prodrug **N6** improved metabolic stability moderately, indicating that it has a good development prospect.

## 4. Experimental Section

### 4.1. Chemistry

^1^H NMR and ^13^C NMR spectra were recorded on Bruker Avance-600 NMR spectrometer with DMSO-*d*_6_ as the solvent. Chemical shifts were expressed in δ values (ppm), and *J* values were expressed in hertz (Hz). Various solvents were obtained from Sinopharm Chemical Reagent Co., Ltd. (SCRC, Shanghai, China), which were of AR grade. Reagents were purchased from Bide Pharmatech Co., Ltd. TLC was performed on silica gel GF254 (Merck) and irradiated by ultraviolet light (*λ* = 254 nm). Flash column chromatography was performed on a column packed with Silica Gel60 (200–300 mesh). Spectra of all compounds are in Appendix A.

#### 4.1.1. 4-Fluoro-3-((4-hydroxypiperidin-1-yl)sulfonyl)benzoic Acid (**N2**)

4-hydroxypiperidine (10.00 mmol, 1.01 g) was dissolved in 10 mL anhydrous acetonitrile, and *N*, *N*-diisopropylethylamine (20.00 mmol, 3.48 mL) was added. At 0 °C, 5 mL acetonitrile solution containing 3-chlorosulfonyl-4-fluorobenzoic acid (10.00 mmol, 2.38 g) was slowly dropped and stirred for 8 h. After the reaction was completed, 10 mL 1N hydrochloric acid solution was added after the solvent was evaporated under reduced pressure. Then, it was extracted with ethyl acetate (3 × 8 mL), combining organic phase, and washed with saturated sodium chloride solution (3 × 15 mL). The solvent was dried by anhydrous sodium sulfate, filtered, and evaporated under reduced pressure to obtain **N2**. White solid, yield: 60.0%. EI-MS: 302.5 [M − H]^−^, C_12_H_14_FNO_5_S (303.30).

#### 4.1.2. 4-Fluoro-3-((4-hydroxypiperidin-1-yl)sulfonyl)-N-(3,4,5-trifluorophenyl)benzamide (**N3**)

**N2** (6.00 mmol, 1.82 g), *O*-(7-azabenzotriazol-1-yl)-*N*,*N*,*N*′,*N*′-tetramethylurea hexafluorophosphate (9.00 mmol, 3.43 g) was dissolved in 15 mL dichloromethane and stirred at room temperature for 1 h. Then, *N*,*N*-diisopropylethylamine (18.00 mmol, 3.13 mL) and 3,4,5-trifluoroaniline (6.00 mmol, 0.88 g) were added and stirred at room temperature for 7 h. After the reaction completed, the solvent was evaporated under reduced pressure, then 20 mL saturated sodium bicarbonate solution was added to the residue in the bottle. It was extracted with ethyl acetate (3 × 10 mL), the organic layer was separated, and 20 mL 1N hydrochloric acid solution was added. Then, it was extracted with ethyl ester (3 × 10 mL), combining the organic layers, and washed with saturated sodium chloride solution (3 × 30 mL). The organic phases were dried over anhydrous sodium sulfate, filtered, and the solvent was evaporated to dryness under reduced pressure. White solid, yield: 34.0%. EI-MS: 432.5 [M + H]^+^, C_18_H_16_F_4_N_2_O_4_S (432.39).

#### 4.1.3. Dibenzyl (1-((2-Fluoro-5-((3,4,5-trifluorophenyl)carbamoyl)phenyl)sulfonyl)piperidin-4-yl) Phosphate (**N4**)

**N3** (2.30 mmol, 1.00 g), 1H-imidazole (3.50 mmol, 0.25 g), and dibenzyl diisopropylphosphoramidite (4.60 mmol, 1.56 mL) were dissolved in 20 mL dichloromethane and stirred for four hours, then 3-chlorobenzoic acid (4.60 mmol, 0.79 g) was added at 0 °C and stirred for an additional 6 h. After the completion of the reaction, 20 mL saturated sodium bicarbonate was added, and then dichloromethane (3 × 10 mL) was added for extraction. The organic phases were combined and washed with saturated sodium chloride (3 × 15 mL). The organic phase was dried over anhydrous sodium sulfate and subjected to column chromatography after stirring with silica gel. White solid, yield: 30.0%. EI-MS: 693.06 [M + H]^+^, C_32_H_29_F_4_N_2_O_7_PS (692.14).

#### 4.1.4. 1-((2-Fluoro-5-((3,4,5-trifluorophenyl)carbamoyl)phenyl)sulfonyl)piperidin-4-yl Dihydrogen Phosphate (**N5**)

**N4** (0.20 g) and Palladium carbon (0.02 g) were dissolved in 15 mL tetrahydrofuran and stirred under hydrogen atmosphere for 12 h. After completion of the reaction, the reaction solution was filtered through celite, and the solvent was evaporated to dryness under reduced pressure and purified by high performance liquid chromatography (HPLC). White solid, yield: 26.4%, mp: 162–164 °C. ^1^H NMR (600 MHz, DMSO-*d*_6_) δ 10.79 (s, 1H), 8.33 (ddd, *J* = 20.1, 7.3, 3.0 Hz, 2H), 7.75–7.65 (m, 3H), 4.27 (s, 2H), 4.03 (*J* = 7.6, 3.6 Hz, 1H), 3.15–2.96 (m, 4H), 1.92–1.64 (m, 4H). ^13^C NMR (151 MHz, DMSO-*d*_6_) δ 163.53, 160.89, 159.17, 149.89 (ddd, ^1^*J*_CF_ = 243 Hz, ^2^*J*_CF_ = 6 Hz, ^3^*J*_CF_ = 3 Hz, C × 2), 135.18 (m), 130.98 (d, ^3^*J*_CF_ = 3 Hz), 130.40 (C × 2), 125.21 (d, ^2^*J*_CF_ = 15 Hz), 117.98 (d, ^2^*J*_CF_ = 22.5 Hz), 104.84 (d, ^2^*J*_CF_ = 22.5 Hz, C × 2), 69.26, 42.19 (C × 2), 31.27 (C × 2). EI-MS: 513.08 [M + H]^+^, C_18_H_17_F_4_N_2_O_7_PS (512.04).

#### 4.1.5. General Procedure for the Synthesis of Target Compounds **N6**–**N8**

**N3** (2.01 mmol, 0.87 g), triethylamine (4.02 mmol, 0.56 mL), 4-dimethylaminopyridine (4.02 mmol, 0.49 g), and different acid anhydrides (12.06 mmol) were added to 15 mL anhydrous dichloromethane and stirred at room temperature for 10 h. After the reaction was completed, 20 mL 1N hydrochloric acid solution was added to the reaction solution to quench the reaction and extracted with dichloromethane (3 × 10 mL). The organic layers were combined, washed with saturated sodium chloride solution (3 × 30 mL), and the organic phase was washed with dried-over anhydrous sodium sulfate; it was filtered, evaporated to dryness under reduced pressure, and separated by silica gel column chromatography.

4-((1-((2-fluoro-5-((3,4,5-trifluorophenyl)carbamoyl)phenyl)sulfonyl)piperidin-4-yl)oxy)-4-oxobutanoic acid (**N6**). White solid, yield: 43.9%, mp: 138–140 °C. ^1^H NMR (600 MHz, DMSO-d_6_) δ 12.22 (s, 1H), 10.83 (s, 1H), 8.34 (ddd, *J* = 15.8, 7.6, 3.3 Hz, 2H), 7.84–7.63 (m, 3H), 4.81 (tt, *J* = 7.7, 3.7 Hz, 1H), 3.31 (d, *J* = 4.6 Hz, 2H), 3.15 (ddd, *J* = 12.1, 7.8, 3.5 Hz, 2H), 2.44 (tt, *J* = 6.5, 3.3 Hz, 4H), 1.88 (ddt, *J* = 11.5, 7.2, 3.7 Hz, 2H), 1.61 (ddq, *J* = 12.0, 7.9, 3.8 Hz, 2H). ^13^C NMR (150 MHz, DMSO-d_6_) δ 173.85, 171.79, 164.01, 160.64 (d, ^1^*J*_CF_ = 259.7 Hz), 150.43 (ddd, ^1^*J*_CF_ = 243.6 Hz, ^2^*J*_CF_ = 9.9 Hz, ^3^*J*_CF_ =5.2 Hz, C × 2), 136.23 (dtd, ^1^*J*_CF_ = 136.4 Hz, ^2^*J*_CF_ = 12.1 Hz, ^3^*J*_CF_ = 4.2 Hz), 135.88 (d, ^3^*J*_CF_ = 9.7 Hz), 131.39 (d, ^3^*J*_CF_ = 3.3 Hz), 130.93 (C × 2), 125.78 (d, ^2^*J*_CF_ = 15.7 Hz), 118.62 (d, ^2^*J*_CF_ = 23.0 Hz), 105.34 (d, ^2^*J*_CF_ = 24.4 Hz, C × 2), 68.28, 51.83, 43.04, 30.18, 29.44, 29.14, 28.98. EI-MS: 531.3 [M − H]^−^, C_22_H_20_F_4_N_2_O_7_S (532.46).

5-((1-((2-fluoro-5-((3,4,5-trifluorophenyl)carbamoyl)phenyl)sulfonyl)piperidin-4-yl)oxy)-5-oxopentanoic acid (**N7**). White solid, yield: 42.6%, mp: 135–137 °C. ^1^H NMR (600 MHz, DMSO-d_6_) δ 11.24 (s, 1H), 9.97 (s, 1H), 8.25 (ddd, *J* = 16.8, 8.6, 3.8 Hz 2H), 6.90–6.73 (m, 3H), 4.00 (tt, *J* = 7.8, 3.8 Hz, 1H), 2.74–2.50 (m, 4H), 1.69 (p, *J* = 1.8 Hz, 2H), 1.64–1.12 (m, 4H), 1.15–0.70 (m, 4H). ^13^C NMR (150 MHz, DMSO-d_6_) δ 174.33, 172.20, 164.01, 161.50 (d, ^1^*J*_CF_ = 259.7 Hz), 150 (ddd, ^1^*J*_CF_ = 244.6 Hz, ^2^*J*_CF_ =9 Hz, ^3^*J*_CF_ = 4.5 Hz, C × 2), 135.7 (m), 134.81 (d, ^3^*J*_CF_ = 16.5 Hz), 131.43 (d, ^3^*J*_CF_ = 3.0 Hz), 130.88 (C × 2), 125.95 (d, ^2^*J*_CF_ = 15.0 Hz), 118.65 (d, ^2^*J*_CF_ = 22.65 Hz), 105.39 (d, ^2^*J*_CF_ = 24.1 Hz, C × 2), 68.31, 43.16, 33.33, 33.28, 33.10, 30.31, 20.47, 20.37. EI-MS: 545.3 [M − H]^−^, C_23_H_22_F_4_N_2_O_7_S (546.11).

6-((1-((2-fluoro-5-((3,4,5-trifluorophenyl)carbamoyl)phenyl)sulfonyl)piperidin-4-yl)oxy)-6-oxohexanoic acid (**N8**). White solid, yield: 39.6%, mp: 132–134 °C. ^1^H NMR (600 MHz, DMSO-d_6_) δ 11.16 (s, 1H), 9.97 (s, 1H), 7.97–7.31 (m, 2H), 7.27–5.61 (m, 3H), 4.00 (tt, *J* = 7.6, 3.8 Hz, 1H), 2.56 (ddd, *J* = 11.5, 6.8, 4.0 Hz, 4H), 2.33 (ddd, *J* = 12.2, 8.4, 3.5 Hz, 2H), 1.70 (p, *J* = 1.9 Hz, 2H), 1.52–1.31 (m, 2H), 1.26–0.97 (m, 2H), 0.96–0.70 (m, 2H), 0.68 (ttd, *J* = 7.7, 5.4, 4.6, 2.8 Hz, 2H). ^13^C NMR (150 MHz, DMSO-d_6_) δ 174.07, 171.85, 163.45, 160.08 (d, ^1^*J*_CF_ = 259.7 Hz), 149.90 (ddd, ^1^*J*_CF_ = 241.6 Hz, ^2^*J*_CF_ = 10.5 Hz, ^3^*J*_CF_ = 6.0 Hz, C × 2), 135.23 (m), 134.26 (d, ^3^*J*_CF_ = 16.5 Hz), 130.87 (d, ^3^*J*_CF_ = 4.5 Hz), 130.33 (C × 2), 125.38 (d, ^2^*J*_CF_ = 15.1 Hz), 118.01 (d, ^2^*J*_CF_ = 24.1 Hz), 104.83 (d, ^2^*J*_CF_ = 25.5 Hz, C × 2), 67.67, 42.61, 42.54, 38.15, 33.28, 33.19, 29.76, 23.85, 23.76. EI-MS: 559.27 [M − H]^−^, C_24_H_24_F_4_N_2_O_7_S (560.12).

### 4.2. In Vitro Anti-HBV Assay

#### 4.2.1. Assessment of Inhibitory Activity on HBV Replication in Hep38.7-Tet Cells

Hep38.7-Tet cells were cultured with DMEM/F-12/GlutaMAX supplemented with 10% fetal bovine serum, and 10 mM HEPES, 200 unit/mL penicillin, 200 µg/mL streptomycin, 5 µg/mL insulin, and 400 ng/mL tetracycline were added at 37 °C in 5% CO_2_. Tetracycline was removed when inducing HBV replication. Three days after seeding the cells, the cells were treated with or without compounds for 6 days and the culture supernatants were recovered to quantify HBV DNA using the primers 5′-AAGGTAGGAGCTGGAGCATTCG-3′, 5′-AGGCGGATTTGCTGGCAAAG-3′, and a probe 5′-FAM-AGCCCTCAGGCTCAGGGCATAC-TAMRA-3′ by real-time fluorescence quantitative PCR, as described [26]. We set concentration gradients (1.00 µM, 0.33 µM, and 0.11 µM) to test and calculate EC_50_ of all compounds.

#### 4.2.2. Cytotoxicity Assay

The WST assays were performed by using Cell Counting Kit-8 (CCK-8) (DOJINDO) according to the manufacturer’s protocol. CCK-8 solution was added to the cells, which were incubated at 37 °C for 15 min and the absorbance at 450 nm was measured. CC_50_ was calculated by the concentration gradient (1.0 µM, 4.0 µM, 16.0 µM, 64.0 µM, and 256.0 µM).

### 4.3. Water Solubility Test of ***N6***

The compound was dissolved in DMSO to prepare the mother liquor with a concentration of 10 mg/mL, and then 10 µL mother liquor was added to 1 mL phosphate buffer with different pH (pH = 2.0, 7.0, and 7.4). The solution was oscillated for 2 h at 3000 rpm of the vortex oscillator, and the precipitation of compounds was observed visually. The standard curve can be established by increasing the concentration of mother liquor until turbidity appears. The peak area (A) of different concentrations (c) was determined by HPLC. The standard curve was established with the concentration and peak area as abscissa and ordinate, respectively, and the standard curve equation A = kc + b was calculated.

The standard curve of NVR 3-778 concentration–peak area was established by measuring the peak area at five concentrations of 1000, 200, 40, 8.0, and 1.6 µg/mL. The standard curve equation was A = 35,268c + 80,359 with R^2^ = 0.9999. By measuring the peak area at 4000 µg/mL, 1000 µg/mL, 200 µg/mL, 40 µg/mL, and 8.0 µg/mL, the standard curve of **N6** concentration–peak area was established: A = 16,303c + 714,762, R^2^ = 0.999. Since the solubility of **N6** in phosphate buffer solution (pH = 7.0, 7.4) had exceeded the detector range, the saturated solution of **N6** was diluted 10 times for testing. On this basis, the saturated solution (pH = 2.0, 7.0, and 7.4) of phosphate buffer at different pH values was prepared and fully oscillated for 1 h before filtration. Finally, the absorption peak area of the sample at different pH phosphate buffer (pH = 2.0, 7.0, and 7.4) was determined (measured twice). According to the standard curve equation, the corresponding concentration was calculated, which was the solubility of the compound.

### 4.4. Plasma and Whole Blood Stability Experiments of ***N6***

A total of 1 mL of rat plasma was placed in ice water at 4 °C and 1 mL of rat whole blood was placed in a water bath at 37 °C for use, respectively. DMSO solution with compound concentration of 1 mM was prepared, and then 10 µL compound solution was added to 90 μL 45% methanol aqueous solution to obtain a quantitative solution with a concentration of 100 µM. A 2 µL compound solution was added to 98 µL 4 °C plasma and 37 °C whole blood, and three portions were prepared to reach the final concentration of 2 µM, respectively. After incubation for two hours, 400 µL termination solution was added. The samples were centrifuged at 4000 rpm for 10 min, and the supernatant was extracted. A total of 100 µL supernatant was mixed with 200 µL ultrapure water and fully shaken for 10 min. The absorption peak area (A) of the sample to be tested was determined by HPLC.

### 4.5. Pharmacokinetic Experiment of ***N6***

Six male SD rats were randomly divided into two groups (administered with **N6** and NVR 3-778, respectively). The rats were fasted for 12 h before administration and had free access to water. Animal feed was given 4 h after administration. Compounds **N6** and NVR 3-778 were both administered orally at a dose of 10 mg/kg. The corresponding doses of compounds were dissolved in DMSO/PEG400/water (3%/60%/37%, V/V/V) mixed solution to prepare compound oral solutions, respectively, which were filtered through the 0.2 µM filter. After intragastric administration, blood was obtained by jugular vein puncture (blood point: before administration and 0.25 h, 0.5 h, 1 h, 2 h, 4 h, 6 h, 10 h, and 24 h after administration). Then 0.25 mL whole blood was collected from each rat at each time point, adding EDTA-K2 for anticoagulation. After collection, the plasma was centrifuged within 1 h (centrifugal condition: 8000 rpm, 5 min, 8 °C). Plasma samples were stored in a refrigerator before analysis. After the experiment, all rats were executed according to the regulations of the Experimental Animal Ethics Committee.

## Data Availability

Not applicable.

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
