# Peer review of "Design, Synthesis, and Evaluation of a Set of Carboxylic Acid and Phosphate Prodrugs Derived from HBV Capsid Protein Allosteric Modulator NVR 3-778"

_molecules, 2022, doi:10.3390/molecules27185987_

Round 1
Reviewer 1 Report
The present study by Zhan et al. describes a series of NVR 3-778 prodrugs. Some compounds exhibited promising anti-HBV activities, improved water solubility and stability, favourable PK properties. This is a very timely topic, and this work has important guidance for the further development of NVR 3-778. On the whole, this manuscript is well organized, and can be accepted by Molecules after some revisions.
(1) There are two expressions in the article: core protein and capsid protein. Whether there is a difference between each other. This may cause trouble to readers. Please express it uniformly or explain the problem in the text.
(2) In Section 1, there are few descriptions about the design of prodrug. Please add this part to explain why these substituents are selected
(3) Some errors:
- In Scheme 1, Please put the structure of N5 on the right side of N4 instead of the lower side.
- In Table 1, oxygen atom and substituent R should be connected by chemical bond.
- Please add “CC50 =” in line 102 and 104.
Author Response
Reviewing: 1
The present study by Zhan et al. describes a series of NVR 3-778 prodrugs. Some compounds exhibited promising anti-HBV activities, improved water solubility and stability, favourable PK properties. This is a very timely topic, and this work has important guidance for the further development of NVR 3-778. On the whole, this manuscript is well organized, and can be accepted by Molecules after some revisions.
- There are two expressions in the article: core protein and capsid protein. Whether there is a difference between each other. This may cause trouble to readers. Please express it uniformly or explain the problem in the text.
Thank you for your advice. We have unified it in the manuscript.
- In Section 1, there are few descriptions about the design of prodrug. Please add this part to explain why these substituents are selected.
Thank you for your advice. Related content has been added in the introduction.
(3) Some errors:
- In Scheme 1, Please put the structure of N5 on the right side of N4 instead of the lower side.
Thank you for your advice. We have changed it in the manuscript.
- In Table 1, oxygen atom and substituent R should be connected by chemical bond.
Thank you for your advice. We have changed it in the manuscript.
- Please add “CC50 =” in line 102 and 104.
Thank you for your advice. We have supplemented it in the manuscript.
Reviewer 2 Report
The paper entitled “Design, synthesis, and evaluation of a set of carboxylic acid and phosphate prodrugs derived from HBV capsid protein allosteric modulator NVR 3-778” by Ji et al. describes a series of novel carboxylic acid and phosphate prodrugs that have anti-HBV activity. The water solubility of compound N6 has been greatly improved compared to NVR3-778. In addition, the plasma and whole blood stability of compound N6 was also improved compared to NVR3-778. Some pharmacokinetic parameters were also improved. In general, this study is relatively comprehensive for prodrug research but some revisions still should be done to improve the quality of this manuscript.
Major:
1.In Paragraph 5 of introduction, the author directly transferred from the prodrug strategy to the caboxylic acid and phosphate prodrugs, which lacks an example transition in the middle.
2.I don 't see the mass spectra of N5 and N7 in the supplementary material, please add.
Minor:
1.Page 2, line 47, continuously should be placed behind.
2.Page 2, line 47, become should be replaced by becomes.
3.Page 2, lines 51-52, remove unnecessary and to make sentences more fluent.
4.Line71, strategy should be replaced by a method or approach.
5.Line81, Scheme 1 Should use bold fonts.
6.Line95, reproduce should be replaced with reproduces.
7.Line101, NVR 3-778 should be followed by its CC50.
8.Line122, verify should be replaced by verified.
9.Section 2.5, lines 140-141, plasma concentrations should be replaced by plasma concentration.
10.Line153, N6 should use bold fonts.
11.Line 182, improve should be replaced by improved.
12.Line 184, had should be replaced by has.
13.Line 293, Reference should be introduced.
Author Response
Major:
- In Paragraph 5 of introduction, the author directly transferred from the prodrug strategy to the caboxylic acid and phosphate prodrugs, which lacks an example transition in the middle.
Thank you for your advice. Related content has been added in the introduction.
2.I don 't see the mass spectra of N5 and N7 in the supplementary material, please add.
Thank you for your advice. Both mass spectra have been supplemented in the supplementary document.
Minor:
- Page 2, line 47, continuously should be placed behind.
Thank you for your advice. We 've made changes in the manuscript.
- Page 2, line 47, become should be replaced by becomes.
Thank you for your advice. We 've made changes in the manuscript.
3.Page 2, lines 51-52, remove unnecessary and to make sentences more fluent.
Thank you for your advice. We 've made changes in the manuscript.
- Line71, strategy should be replaced by a method or approach.
Thank you for your advice. We 've made changes in the manuscript.
- Line81, Scheme 1 Should use bold fonts.
Thank you for your advice. We 've made changes in the manuscript.
- Line95, reproduce should be replaced with reproduces.
Thank you for your advice. We 've made changes in the manuscript.
7.Line101, NVR 3-778 should be followed by its CC50.
Thank you for your advice. We 've made changes in the manuscript.
- Line122, verify should be replaced by verified.
Thank you for your advice. We 've made changes in the manuscript.
9.Section 2.5, lines 140-141, plasma concentrations should be replaced by plasma concentration.
Thank you for your advice. We 've made changes in the manuscript.
- Line153, N6 should use bold fonts.
Thank you for your advice. We 've made changes in the manuscript.
11.Line 182, improve should be replaced by improved.
Thank you for your advice. We 've made changes in the manuscript.
- Line 184, had should be replaced by has.
Thank you for your advice. We 've made changes in the manuscript.
13.Line 293, Reference should be introduced.
Thank you for your advice. We 've made changes in the manuscript.
Author Response
Reviewing: 3
Why was N8 not used for aqueous solubility testing and PK studies since it showed better safety margin in vitro?
Thanks for your advice. The study of hemisuccinate prodrugs is more extensive, with Artesunate, Chloramphenicol Hemisuccinate, Hydrocortisone Sodium Succinat, Vitamin E Succinate and other drugs. Therefore, we preliminarily studied the hemisuccinate prodrug N6, and we will further study N8 in the near future.
This is misleading and does not support the rational for this work.The aim of the study was to increase aqueous solubility and hence improve on PK parameters, but this does not seem to be the result. The EC50 values are in micromolars, while the Cmax is in lower units. The question here is, is the Cmax sufficient to maintain efficacy? This has to be discussed.
Thanks for your advice, we 've removed misleading language. NVR3-778 has a narrow therapeutic window and its development was stopped because of a limited efficacy at clinically feasible. In this study, while improving its water solubility, we hope to improve its pharmacokinetic properties to some extent. We successfully improved its half-life and made the drug release more slowly, which is helpful to improve the therapeutic window to a certain extent.
Minor:
Thank you for your advice, some small questions we have revised in the manuscript.
Round 2
Reviewer 3 Report
N/A